# CONSISTENCY REGULARIZATION
# FOR GENERATIVE ADVERSARIAL NETWORKS

**Han Zhang, Zizhao Zhang, Augustus Odena, Honglak Lee**
Google Research
`{zhanghan,zizhaoz,augustusodena,honglak}@google.com`

## ABSTRACT

Generative Adversarial Networks (GANs) are known to be difficult to train, despite considerable research effort. Several regularization techniques for stabilizing training have been proposed, but they introduce non-trivial computational overheads and interact poorly with existing techniques like spectral normalization. In this work, we propose a simple, effective training stabilizer based on the notion of consistency regularization—a popular technique in the semi-supervised learning literature. In particular, we augment data passing into the GAN discriminator and penalize the sensitivity of the discriminator to these augmentations. We conduct a series of experiments to demonstrate that consistency regularization works effectively with spectral normalization and various GAN architectures, loss functions and optimizer settings. Our method achieves the best FID scores for unconditional image generation compared to other regularization methods on CIFAR-10 and CelebA. Moreover, Our consistency regularized GAN (CR-GAN) improves state-of-the-art FID scores for conditional generation from 14.73 to 11.48 on CIFAR-10 and from 8.73 to 6.66 on ImageNet-2012.

## 1 INTRODUCTION

Generative Adversarial Networks (GANs) (Goodfellow et al., 2014) have recently demonstrated impressive results on image-synthesis benchmarks (Radford et al., 2016; Zhang et al., 2017; Miyato & Koyama, 2018; Zhang et al., 2018; Brock et al., 2018; Karras et al., 2019). In the original setting, GANs are composed of two neural networks trained with competing goals: the *generator* is trained to synthesize realistic samples to fool the discriminator and the *discriminator* is trained to distinguish real samples from fake ones produced by the generator.

One major problem with GANs is the instability of the training procedure and the general sensitivity of the results to various hyperparameters (Salimans et al., 2016). Because GAN training implicitly requires finding the Nash equilibrium of a non-convex game in a continuous and high dimensional parameter space, it is substantially more complicated than standard neural network training. In fact, formally characterizing the convergence properties of the GAN training procedure is mostly an open problem (Odena, 2019). Previous work (Arjovsky & Bottou, 2017; Miyato et al., 2018a; Odena et al., 2017; Chen et al., 2019; Wei et al., 2018) has shown that interventions focused on the discriminator can mitigate stability issues. Most successful interventions fall into two categories, normalization and regularization. Spectral normalization is the most effective normalization method, in which weight matrices in the discriminator are divided by an approximation of their largest singular value. For regularization, Gulrajani et al. (2017) penalize the gradient norm of straight lines between real data and generated data. Roth et al. (2017) propose to directly regularize the squared gradient norm for both the training data and the generated data. DRAGAN (Kodali et al., 2017) introduces another form of gradient penalty where the gradients at Gaussian perturbations of training data are penalized. One may anticipate simultaneous regularization and normalization could improve sample quality. However, most of these gradient based regularization methods either provide marginal gains or fail to introduce any improvement when normalization is used (Kurach et al., 2019), which is also observed in our experiments. These regularization methods and spectral normalization are motivated by controlling Lipschitz constant of the discriminator. We suspect this might be the reason that applying both does not lead to overlaid gain.

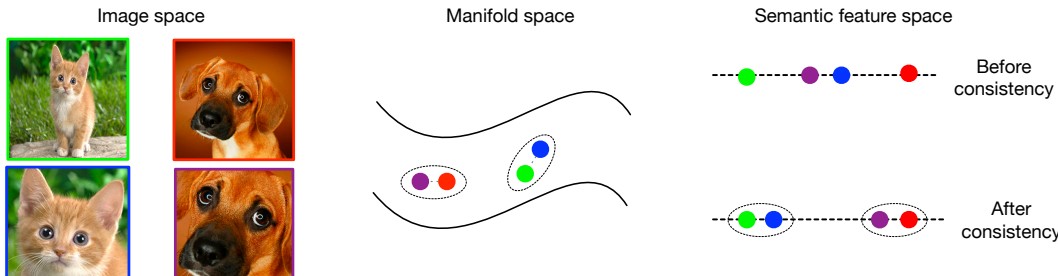

Figure 1: An illustration of consistency regularization for GANs. Before consistency regularization, the zoomed-in dog and the zoomed-in cat (bottom left) can be closer than they are to their original images in feature space induced by the GAN discriminator. This is illustrated in the upper right (the semantic feature space), where the purple dot is closer to the blue dot than to the red dot, and so forth. After we enforce consistency regularization based on the implicit assumption that image augmentation preserves the semantics we care about, the purple dot pulled closer to the red dot.

In this paper, we examine a technique called consistency regularization (Bachman et al., 2014; Sajjadi et al., 2016; Laine & Aila, 2016; Zhai et al., 2019; Xie et al., 2019; Hu et al., 2017) in contrast to gradient-based regularizers. Consistency regularization is widely used in semi-supervised learning to ensure that the classifier output remains unaffected for an unlabeled example even it is augmented in semantic-preserving ways. In light of this intuition, we hypothesize a well-trained discriminator should also be regularized to have the consistency property, which enforces the discriminator to be unchanged by arbitrary semantic-preserving perturbations and to focus more on semantic and structural changes between real and fake data. Therefore, we propose a simple regularizer to the discriminator of GAN: we augment images with semantic-preserving augmentations before they are fed into the GAN discriminator and penalize the sensitivity of the discriminator to those augmentations.

This technique is simple to use and surprisingly effective. It is as well less computationally expensive than prior techniques. More importantly, in our experiments, consistency regularization can always further improve the model performance when spectral normalization is used, whereas the performance gains of previous regularization methods diminish in such case. In extensive ablation studies, we show that it works across a large range of GAN variants and datasets. We also show that simply applying this technique on top of existing GAN models leads to new state-of-the-art results as measured by Frechet Inception Distance (Heusel et al., 2017).

In summary, our contributions are summarized as follows:

- We propose consistency regularization for GAN discriminators to yield a simple, effective regularizer with lower computational cost than gradient-based regularization methods.
- We conduct extensive experiments with different GAN variants to demonstrate that our technique interacts effectively with spectral normalization. Our consistency regularized GAN (CR-GAN) achieves the best FID scores for unconditional image generation on both CIFAR-10 and CelebA.
- We show that simply applying the proposed technique can further boost the performance of state-of-the-art GAN models. We improve FID scores for conditional image generation from 14.73 to 11.48 on CIFAR-10 and from 8.73 to 6.66 on ImageNet-2012.

## 2 METHOD

### 2.1 GANS

A GAN consists of a generator network and a discriminator network. The generator $G$ takes a latent variable $z \sim p(z)$ sampled from a prior distribution and maps it to the observation space $\mathcal{X}$. The discriminator $D$ takes an observation $x \in \mathcal{X}$ and produces a decision output over possible observation sources (either from $G$ or from the empirical data distribution). In the standard GAN training procedure the generator $G$ and the discriminator $D$ are trained by minimizing the following

objectives in an alternating fashion:

$$L_D = -\mathbb{E}_{x \sim p_{\text{data}}} [\log D(x)] - \mathbb{E}_{z \sim p(z)} [1 - \log D(G(z))],$$
$$L_G = -\mathbb{E}_{z \sim p(z)} [\log D(G(z))],$$

(1)

where $p(z)$ is usually a standard normal distribution. This formulation is originally proposed by Goodfellow et al. (2014) as non-saturating (NS) GAN. A significant amount of research has been done on modifying this formulation in order to improve the training process. A notable example is the hinge-loss version of the adversarial loss (Lim & Ye, 2017; Tran et al., 2017):

$$L_D = -\mathbb{E}_{x \sim p_{\text{data}}} [\min(0, -1 + D(x))] - \mathbb{E}_{z \sim p(z)} [\min(0, -1 - D(G(z)))],$$
$$L_G = -\mathbb{E}_{z \sim p(z)} [D(G(z))].$$

(2)

Another commonly adopted GAN formulation is the Wassertein GAN (WGAN) (Arjovsky et al., 2017), in which the authors propose clipping the weights of the discriminator in an attempt to enforce that the GAN training procedure implicitly optimizes a bound on the Wasserstein distance between the target distribution and the distribution given by the generator. The loss function of WGAN can be written as

$$L_D = -\mathbb{E}_{x \sim p_{\text{data}}} [D(x)] + \mathbb{E}_{z \sim p(z)} [D(G(z))],$$
$$L_G = -\mathbb{E}_{z \sim p(z)} [D(G(z))].$$

(3)

Subsequent work has refined this technique in several ways (Gulrajani et al., 2017; Miyato et al., 2018a; Zhang et al., 2019), and the current widely-used practice is to enforce spectral normalization (Miyato et al., 2018a) on both the generator and the discriminator.

## 2.2 CONSISTENCY REGULARIZATION

Consistency regularization has emerged as a gold-standard technique (Sajjadi et al., 2016; Laine & Aila, 2016; Zhai et al., 2019; Xie et al., 2019; Oliver et al., 2018; Berthelot et al., 2019) for semi-supervised learning on image data. The basic idea is simple: an input image is perturbed in some semantics-preserving ways and the sensitivity of the classifier to that perturbation is penalized. The perturbation can take many forms: it can be image flipping, or cropping, or adversarial attacks. The regularization form is either the mean-squared-error (Sajjadi et al., 2016; Laine & Aila, 2016) between the model's output for a perturbed and non-perturbed input or the KL divergence (Xie et al., 2019; Miyato et al., 2018b) between the distribution over classes implied by the output logits.

## 2.3 CONSISTENCY REGULARIZATION FOR GANS

The goal of the discriminator in GANs is to distinguish real data from fake ones produced by the generator. The decision should be invariant to any valid domain-specific data augmentations. For example, in the image domain, the image being real or not should not change if we flip the image horizontally or translate the image by a few pixels. However, the discriminator in GANs does not guarantee this property explicitly.

To resolve this, we propose a consistency regularization on the GAN discriminator during training. In practice, we randomly augment training images as they are passed to the discriminator and penalize the sensitivity of the discriminator to those augmentations.

We use $D_j(x)$ to denote the output vector before activation of the $j$th layer of the discriminator given input $x$. $T(x)$ denotes a stochastic data augmentation function. This function can be linear or nonlinear, but aims to preserve the semantics of the input. Our proposed regularization is given by

$$\min_D \ L_{cr} = \ \min_D \sum_{j=m}^{n} \lambda_j \big\| D_j(x) - D_j(T(x)) \big\|^2,$$

(4)

where $j$ indexes the layers, $m$ is the starting layer and $n$ is the ending layer that consistency is enforced. $\lambda_j$ is weight coefficient for $j$th layer and $\|\cdot\|$ denotes $L^2$ norm of a given vector. This consistency regularization encourages the discriminator to produce the same output for a data point under various data augmentations.

---

**Algorithm 1** Consistency Regularized GAN (CR-GAN). We use $\lambda = 10$ by default.

---

**Input:** generator and discriminator parameters $\theta_G, \theta_D$, consistency regularization coefficient $\lambda$, Adam hyperparameters $\alpha, \beta_1, \beta_2$, batch size $M$, number of discriminator iterations per generator iteration $N_D$

1: **for** number of training iterations **do**
2:     **for** $t = 1, ..., N_D$ **do**
3:         **for** $i = 1, ..., M$ **do**
4:             Sample $z \sim p(z)$, $x \sim p_{\text{data}}(x)$
5:             Augment $x$ to get $T(x)$
6:             $L_{cr}^{(i)} \leftarrow \left\| D(x) - D(T(x)) \right\|^2$
7:             $L_D^{(i)} \leftarrow D(G(z)) - D(x)$
8:         **end for**
9:         $\theta_D \leftarrow \text{Adam}(\frac{1}{M} \sum_{i=1}^{M} (L_D^{(i)} + \lambda L_{cr}^{(i)}), \alpha, \beta_1, \beta_2)$
10:     **end for**
11:     Sample a batch of latent variables $\{z^{(i)}\}_{i=1}^{M} \sim p(z)$
12:     $\theta_G \leftarrow \text{Adam}(\frac{1}{M} \sum_{i=1}^{M} (-D(G(z))), \alpha, \beta_1, \beta_2)$
13: **end for**

---

In our experiments, we find that consistency regularization on the last layer of the discriminator before the activation function is sufficient. $L_{cr}$ can be rewritten as

$$L_{cr} = \left\| D(x) - D(T(x)) \right\|^2, \tag{5}$$

where from now on we will drop the layer index for brevity. This cost is added to the discriminator loss (weighted by a hyper-parameter $\lambda$) when updating the discriminator parameters. The generator update remains unchanged. Thus, the overall consistency regularized GAN (CR-GAN) objective is written as

$$L_D^{cr} = L_D + \lambda L_{cr}, \qquad L_G^{cr} = L_G. \tag{6}$$

Our design of $L_{cr}$ is general-purpose and thereby can work with any valid adversarial losses $L_G$ and $L_D$ for GANs (See Section 2.1 for examples). Algorithm 1 illustrates the details of CR-GAN with Wassertein loss as an example. In contrast to previous regularizers, our method does not increase much overhead. The only extra computational cost comes from feeding an additional (third) image through the discriminator forward and backward when updating the discriminator parameters.

## 3 EXPERIMENTS

This section validates our proposed CR-GAN method. First we conduct a large scale study to compare consistency regularization to existing GAN regularization techniques (Kodali et al., 2017; Gulrajani et al., 2017; Roth et al., 2017) for several GAN architectures, loss functions and other hyperparameter settings. We then apply consistency regularization to a state-of-the-art GAN model (Brock et al., 2018) and demonstrate performance improvement. Finally, we conduct ablation studies to investigate the importance of various design choices and hyper-parameters. All our experiments are based on the open-source code from Compare GAN (Kurach et al., 2019), which is available at https://github.com/google/compare_gan.

### 3.1 DATASETS AND EVALUATION METRICS

We validate our proposed method on three datasets: CIFAR-10 (Krizhevsky, 2009), CELEBA-HQ-128 (Karras et al., 2018), and ImageNet-2012 (Russakovsky et al., 2015). We follow the procedure in Kurach et al. (2019) to prepare datasets. CIFAR-10 consists of 60K of $32 \times 32$ images in 10 classes; 50K for training and 10K for testing. CELEBA-HQ-128 (CelebA) contains 30K images of faces at a resolution of $128 \times 128$. We use 3K images for testing and the rest of images for training. ImageNet-2012 contains roughly 1.2 million images with 1000 distinct categories and we down-sample the images to $128 \times 128$ in our experiments.

We adopt the Fréchet Inception distance (FID) (Heusel et al., 2017) as primitive metric for quantitative evaluation, as FID has proved be more consistent with human evaluation. In our experiments the

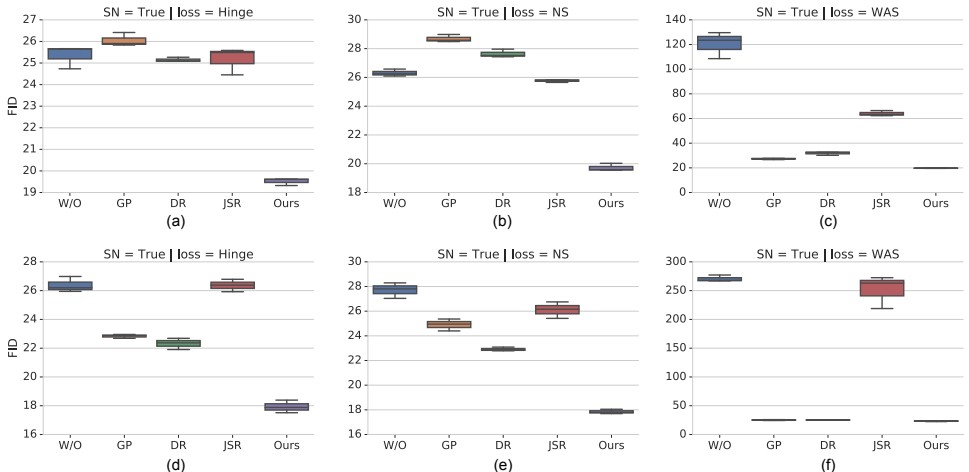

Figure 2: Comparison of our method with existing regularization techniques under different GAN losses. Techniques include no regularization (W/O), Gradient Penalty (GP) (Gulrajani et al., 2017), DRAGAN (DR) (Kodali et al., 2017) and JS-Regularizer (JSR) (Roth et al., 2017). Results (a-c) are for CIFAR-10 and results (d-f) are for CelebA.

FID is calculated on the test dataset. In particular, we use 10K generated images vs. 10K test images on CIFAR-10, 3K vs. 3K on CelebA and 50K vs. 50K on ImageNet. We also provide the Inception Score (Salimans et al., 2016) for different methods in the Appendix F for supplementary results. By default, the augmentation used in consistency regularization is a combination of randomly shifting the image by a few pixels and randomly flipping the image horizontally. The shift size is 4 pixels for CIFAR-10 and CelebA and 16 for ImageNet.

## 3.2 COMPARISON WITH OTHER GAN REGULARIZATION METHODS

In this section, we compare our methods with three GAN regularization techniques, Gradient Penalty (GP) (Gulrajani et al., 2017), DRAGAN Regularizer (DR) (Kodali et al., 2017) and JS-Regularizer (JSR) (Roth et al., 2017) on CIFAR-10 and CelebA.

Following the procedures from (Kurach et al., 2019; Lucic et al., 2018), we evaluate these methods across different optimizer parameters, loss functions, regularization coefficient and neural architectures. For optimization, we use the Adam optimizer with batch size of 64 for all our experiments. We stop training after 200k generator update steps for CIFAR-10 and 100k steps for CelebA. By default, spectral normalization (SN) (Miyato et al., 2018a) is used in the discriminator, as this is the most effective normalization method for GANs (Kurach et al., 2019) and is becoming the standard for 'modern' GANs (Zhang et al., 2019; Brock et al., 2018). Results without spectral normalization can be seen in the Appendix B.

### 3.2.1 IMPACT OF LOSS FUNCTION

In this section, we discuss how each regularization method performs when the loss function is changed. Specifically, we evaluate regularization methods using three loss functions: the non-saturating loss (NS) (Goodfellow et al., 2014), the Wasserstein loss (WAS) (Arjovsky et al., 2017), and the hinge loss (Hinge) (Lim & Ye, 2017; Tran et al., 2017). For each loss function, we evaluate over 7 hyper-parameter settings of the Adam optimizer (more details in Section A of the appendix). For each configuration, we run each model 3 times with different random seeds. For the regularization coefficient, we use the best value reported in the corresponding paper. Specifically $\lambda$ is set to be 10 for both GP, DR and our method and 0.1 for JSR. In this experiment, we use the SNDCGAN network architecture (Miyato et al., 2018a) for simplicity. In the end, similar as Kurach et al. (2019), we aggregate all runs and report the FID distribution of the top 15% of trained models.

The results are shown in Figure 2. The consistency regularization improves the baseline across all different loss functions and both datasets. Other techniques have more mixed results: For example,

| Setting | W/O | GP | DR | JSR | Ours (CR-GAN) |
|---|---|---|---|---|---|
| CIFAR-10 (SNDCGAN) | 24.73 | 25.83 | 25.08 | 25.17 | **18.72** |
| CIFAR-10 (ResNet) | 19.00 | 19.74 | 18.94 | 19.59 | **14.56** |
| CelebA (SNDCGAN) | 25.95 | 22.57 | 21.91 | 22.17 | **16.97** |

Table 1: Best FID scores for unconditional image generation on CIFAR-10 and CelebA.

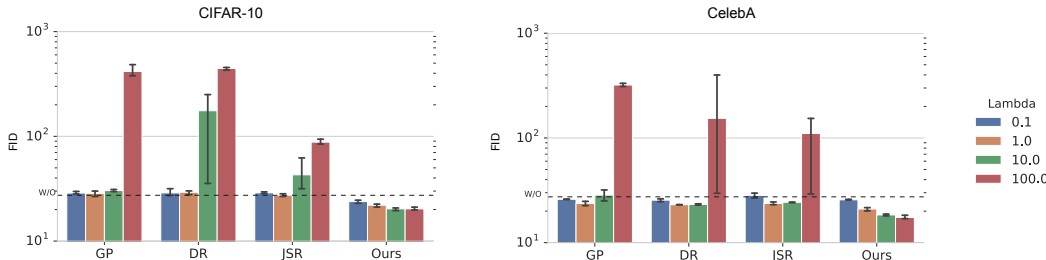

Figure 3: Comparison of FID scores with different values of the regularization coefficient $\lambda$ on CIFAR-10 and CelebA. The dotted line is a model without regularization.

GP and DR can marginally improve the performance for settings (d) and (e) but lead to worse results for settings (a) and (b) (which is consistent with findings from Kurach et al. (2019)). In all cases, our consistency-regularized GAN models have the lowest (best) FID.

This finding is especially encouraging, considering that the consistency regularization has lower computational cost (and is simpler to implement) than the other techniques. In our experiments, the consistency regularization is around 1.7 times faster than gradient based regularization techniques, including DR, GP and JSR, which need to compute the gradient of the gradient norm $\|\nabla_x(D)\|$. Please see Table C1 in the appendix for the actual training speed.

### 3.2.2 IMPACT OF THE REGULARIZATION COEFFICIENT

Here we study the sensitivity of GAN regularization techniques to the regularization coefficient $\lambda$. We train SNDCGANs with non-saturating losses and fix the other hyper-parameters. $\lambda$ is chosen among $\{0.1, 1, 10, 100\}$. The results are shown in Figure 3. From this figure, we can see consistency regularization is more robust to changes in $\lambda$ than other GAN regularization techniques (it also has the best FID for both datasets). The results indicate that consistency regularization can be used as a plug-and-play technique to improve GAN performance in different settings without much hyper-parameter tuning.

### 3.2.3 IMPACT OF NEURAL ARCHITECTURES

To validate whether the above findings hold across different neural architectures, we conduct experiments on CIFAR-10 using a ResNet (He et al., 2016; Gulrajani et al., 2017) architecture instead of an SNDCGAN. All other experimental settings are same as in Section 3.2.1. The FID values are presented in Figure 4. By comparing results in Figure 4 and Figure 2, we can see that results on SNDCGAN and results on ResNet are comparable, though consistency regularization favors even better in this case: In sub-plot (c) of Figure 4, we can see that consistency regularization is the only regularization method that can generate satisfactory samples with a reasonable FID score (The FID scores for other methods are above 100). Please see Figure D3 for the actual generated samples in this setting. As in Section 3.2.1, consistency regularization has the best FID for each setting.

In Table 1, we show FID scores for the best-case settings from this section. Consistency regularization improves on the baseline by a large margin and achieves the best results across different network architectures and datasets. In particular, it achieves an FID 14.56 on CIFAR-10 16.97 on CelebA. In fact, our FID score of 14.56 on CIFAR-10 for *unconditional* image generation is even lower than the 14.73 reported in Brock et al. (2018) for *class-conditional* image-synthesis with a much larger network architecture and much bigger batch size.

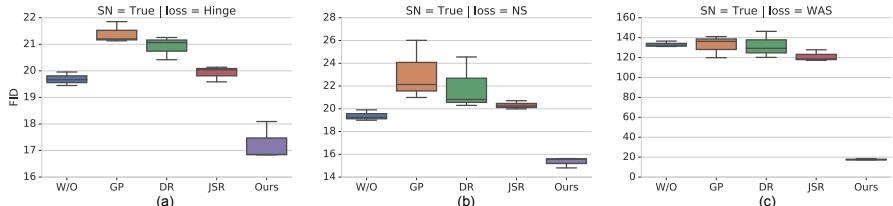

Figure 4: Comparison of FID scores with ResNet structure on different loss settings on CIFAR-10.

### 3.3 COMPARISON WITH STATE-OF-THE-ART GAN MODELS

In this section, we add consistency regularization to the state-of-the-art BigGAN model (Brock et al., 2018) and perform class conditional image-synthesis on CIFAR-10 and ImageNet. Our model has exactly the same architecture and is trained under the same settings as BigGAN⋆, the open-source implementation of BigGAN from Kurach et al. (2019). The only difference is that our model uses consistency regularization. In Table 2, we report the original FID scores without noise truncation. Consistency regularization improves the FID score of BigGAN⋆ on CIFAR-10 from 20.42 to 11.48. In addition, the FID on ImageNet is improved from 7.75 to 6.66.

Generated samples for CIFAR-10 and ImageNet with consistency regularized models and baseline models are shown in Figures E1, E2 and E3 in the appendix.

| Dataset | SNGAN | SAGAN | BigGAN | BigGAN⋆ | CR-BigGAN⋆ |
|---|---|---|---|---|---|
| CIFAR-10 | 17.5 | / | 14.73 | 20.42 | **11.48** |
| ImageNet | 27.62 | 18.65 | 8.73 | 7.75 | **6.66** |

Table 2: Comparison of our technique with state-of-the-art GAN models including SNGAN (Miyato & Koyama, 2018), SAGAN (Zhang et al., 2019) and BigGAN (Brock et al., 2018) for class conditional image generation on CIFAR-10 and ImageNet in terms of FID. BigGAN⋆ is the Big-GAN implementation of Kurach et al. (2019). CR-BigGAN⋆ has the exactly same architecture as BigGAN⋆ and is trained with the same settings. The only difference is CR-BigGAN⋆ adds consistency regularization.

## 4 ABLATION STUDIES AND DISCUSSION

### 4.1 HOW MUCH DOES AUGMENTATION MATTER BY ITSELF?

Our consistency regularization technique actually has two parts: we perform data augmentation on inputs from the training data, and then consistency is enforced between the augmented data and the original data. We are interested in whether the performance gains shown in Section 3 are merely due to data augmentation, since data augmentation reduces the over-fitting of the discriminator to the input data. Therefore, we have designed an experiment to answer this question. First, we train three GANs: (1) a GAN trained with consistency regularization, as in Algorithm 1, (2) a baseline GAN trained without augmentation or consistency regularization, and (3) a GAN trained with only data augmentation and no consistency regularization. We then plot (Figure 5) both their FID and the test accuracy of their discriminator on a held-out test set. The FID tells us how 'good' the resulting GAN is, and the discriminator test accuracy tells us how much the GAN discriminator over-fits. Interestingly, we find that these two measures are not well correlated in this case. The model trained with only data augmentation over-fits substantially less than the baseline GAN, but has almost the same FID. The model trained with consistency regularization has the same amount of over-fitting as the model trained with just data augmentation, but a much lower FID.

This suggests an interesting hypothesis, which is that the mechanism by which the consistency regularization improves GANs is not simply discriminator generalization (in terms of classifying images into real vs fake). We believe that the main reason for the impressive gain from the consistency regularization is due to learning more semantically meaningful representation for the discriminator. More specifically, data augmentation will simply treat all real images and their transformed images

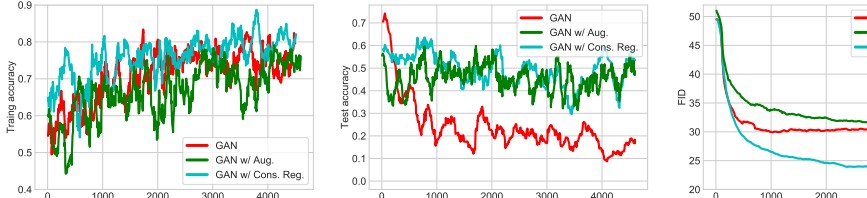

Figure 5: A study of how much data augmentation matters by itself. Three GANs were trained on CIFAR-10: one baseline GAN, one GAN with data augmentation only, and one GAN with consistency regularization. (**Left**) Training accuracy of the GAN discriminator. (**Middle**) Test accuracy of the GAN discriminator on the held out test set. The accuracy is low for the baseline GAN, which indicates it suffered from over-fitting. The accuracy for the other two is basically indistinguishable for each other. This suggests that augmentation by itself is enough to reduce discriminator over-fitting, and that consistency regularization by itself does little to address over-fitting. (**Right**) FID scores of the three settings. The score for the GAN with only augmentation is not any better than the score for the baseline, even though its discriminator is not over-fitting. The score for the GAN with consistency regularization is better than both of the others, suggesting that the consistency regularization acts on the score through some mechanism other than by reducing discriminator over-fitting.

| Metric | Gaussian Noise | Random shift & flip | Cutout | Cutout w/ random shift & flip |
|--------|----------------|---------------------|--------|-------------------------------|
| FID | 21.91±0.32 | 16.04±0.17 | 17.10±0.29 | 19.46±0.26 |

Table 3: FID scores on CIFAR-10 for different types of image augmentation. Gaussian noise is the worst, and random shift and flip is the best, consistent with general consensus on the best way to perform image optimization on CIFAR-10 (Zagoruyko & Komodakis, 2016).

with the same label as real without considering semantics, whereas our consistency regularization further enforces learning implicit manifold structure in the discriminator that pulls semantically similar images (i.e., original real image and the transformed image) to be closer in the discriminator representation space.

## 4.2 How does the Type of Augmentation Affect Results?

To analyze how different types of data augmentation affect our results, we conduct an ablation study on the CIFAR-10 dataset comparing the results of using four different types of image augmentation: (1) adding Gaussian noise to the image in pixel-space, (2) randomly shifting the image by a few pixels and randomly flipping it horizontally, (3) applying cutout (DeVries & Taylor, 2017) transformations to the image, and (4) cutout *and* random shifting and flipping. As shown in Table 3, random flipping and shifting *without* cutout gives the best results (FID 16.04) among all four methods. Adding Gaussian noise in pixel-space gives the worst results. This result empirically suggests that adding Gaussian noise is not a good semantic preserving transformation in the image manifold. It's also noteworthy that the most extensive augmentation (random flipping and shifting with cutout) did not perform the best. One possible reason is that the generator sometimes also generates samples with augmented artifacts (e.g., cutout). If such artifacts do not exist in the real dataset, it might lead to worse FID performance.

## 5 Conclusion

In this paper, we propose a simple, effective, and computationally cheap method – consistency regularization – to improve the performance of GANs. Consistency regularization is compatible with spectral normalization and results in improvements in all of the many contexts in which we evaluated it. Moreover, we have demonstrated consistency regularization is more effective than other regularization methods under different loss functions, neural architectures and optimizer hyper-parameter settings. We have also shown simply applying consistency regularization on top of state-of-the-art GAN models can further greatly boost the performance. Finally, we have conducted a thorough study on the design choices and hyper-parameters of consistency regularization.

## ACKNOWLEDGMENTS

We thank Colin Raffel for feedback on drafts of this article. We also thank Marvin Ritter, Michael Tschannen and Mario Lucic for answering our questions of using compare GAN codebase for large scale GAN evaluation.

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

# APPENDIX

## A  HYPERPARAMETER SETTINGS OF OPTIMIZER

| Setting | $lr$ | $\beta_1$ | $\beta_2$ | $N_{dis}$ |
|---------|------|-----------|-----------|-----------|
| A | 0.0001 | 0.5 | 0.9 | 5 |
| B | 0.0001 | 0.5 | 0.999 | 1 |
| C | 0.0002 | 0.5 | 0.999 | 1 |
| D | 0.0002 | 0.5 | 0.999 | 5 |
| E | 0.001 | 0.5 | 0.9 | 5 |
| F | 0.001 | 0.5 | 0.999 | 5 |
| G | 0.001 | 0.9 | 0.999 | 5 |

Table A1: Hyper-parameters of the optimizer used in our experiments.

Here, similar as the experiments in Miyato et al. (2018a); Kurach et al. (2019), we evaluate all regularization methods across 7 different hyperparameters settings for (1) learning rate $lr$ (2) first and second order momentum parameters of Adam $\beta_1$, $\beta_2$ (3) number of the updates of the discriminator per generator update, $N_{dis}$. The details of all the settings are shown in Table A1. Among all these 7 settings, A-D are the "good" hyperparameters used in previous publications (Radford et al., 2016; Gulrajani et al., 2017; Kurach et al., 2019); E, F, G are the "aggressive" hyperparameter settings introduced by Miyato et al. (2018a) to test model performance under noticeably large learning rate or disruptively high momentum. In practice, we find setting C generally works the best for SNDCGAN and setting D is the optimal setting for ResNet. These two settings are also the default settings in the Compare GAN codebase for the corresponding network architectures.

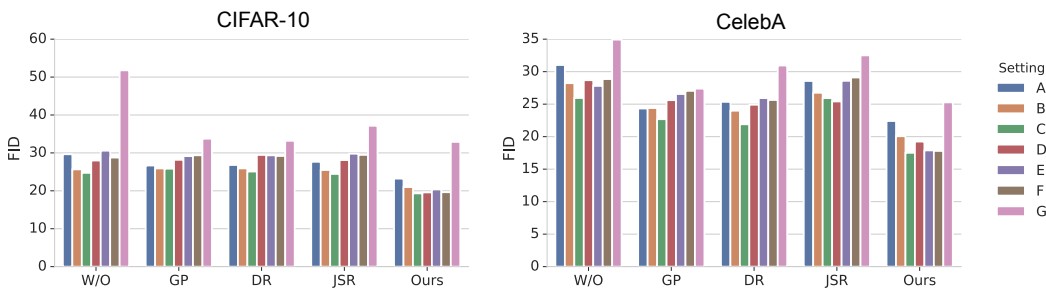

Figure A1: Comparison of FID scores with different optimizer settings.

Figure A1 displays the FID score of all methods with 7 settings A-G. We can observe that consistency regularization is fairly robust even for some of the aggressive hyperparameter settings. In general, the proposed consistency regularization can generate better samples with different optimizer settings compared with other regularization methods.

# B COMPARISON OF DIFFERENT REGULARIZATION METHODS WHEN SPECTRAL NORMALIZATION IS NOT USED

Figure B1: Comparison of FID scores when SN is not used.

Here, we compare different regularization methods when spectral normalization (SN) is not used. As shown in Figure B1, our consistency regularization always improves the baseline model (W/O). It also achieves the best FID scores in most of the cases, which demonstrates that consistency regularization does not depend on spectral normalization. By comparing with the results in Figure 2 and Figure 4, we find adding spectral normalization will further boost the results. More importantly, the consistency regularization is only method that improve on top of spectral normalization without exception. The other regularization methods do not have this property.

# C TRAINING SPEED

Here we show the actual training speed of discriminator updates for SNDCGAN on CIFAR-10 with NVIDIA Tesla V100. Consistency regularization is around 1.7 times faster than gradient based regularization techniques.

| Method | W/O | GP | DR | JSR | Ours (CR-GAN) |
|---|---|---|---|---|---|
| Speed (step/s) | 66.3 | 29.7 | 29.8 | 29.2 | 51.7 |

Table C1: Training speed of discriminator updates for SNDCGAN on CIFAR-10.

## D GENERATED SAMPLES FOR UNCONDITIONAL IMAGE GENERATION

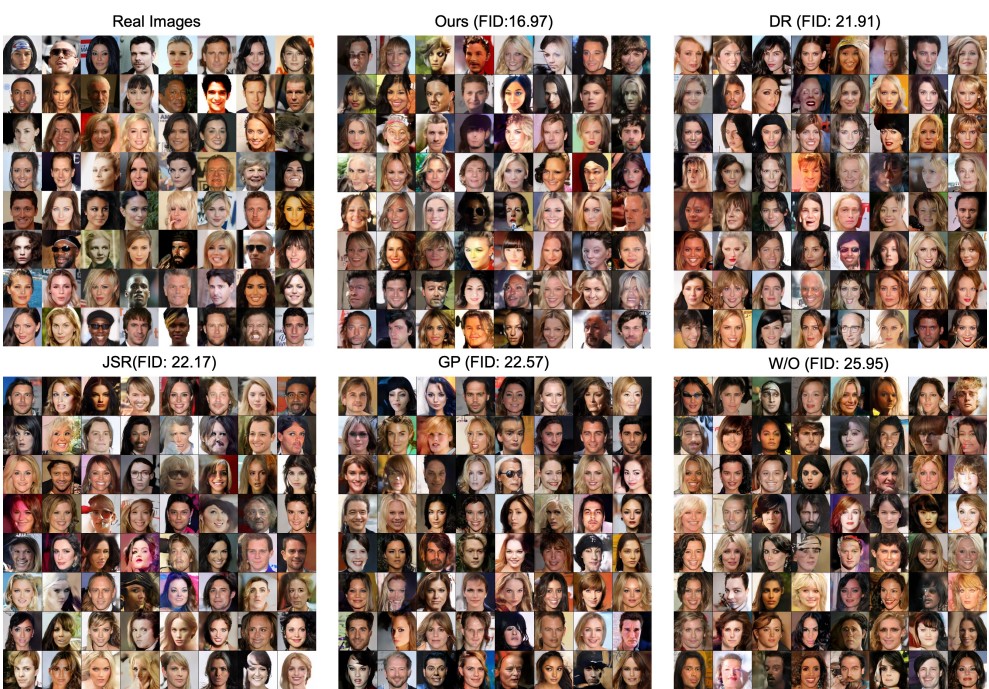

Figure D1: Comparison of generated samples of CelebA.

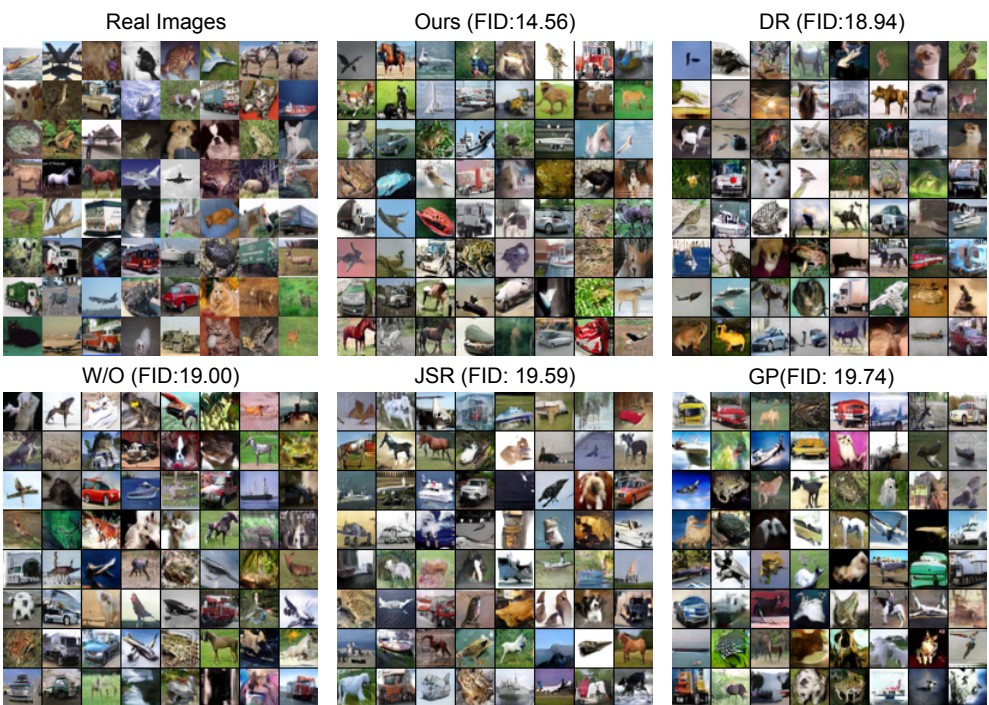

Figure D2: Comparison of generated samples for unconditional image generation on CIFAR-10 with a ResNet architecture.

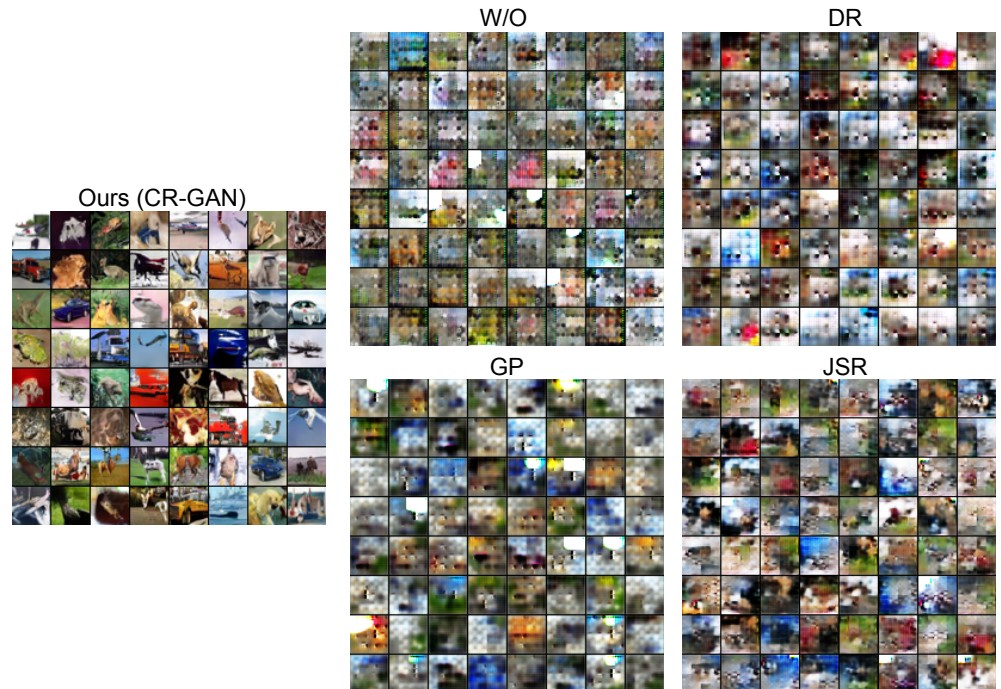

Figure D3: Comparison of unconditional generated samples on CIFAR-10 with a ResNet architecture, Wasserstein loss and spectral normalization. This is a hard hyperparameter setting where the baseline and previous regularization methods fail to generate reasonable samples. Consistency Regularization is the only regularization method that can generate satisfactory samples in this setting. FID scores are shown in sub-plot (c) of Figure 4.

# E    GENERATED SAMPLES FOR CONDITIONAL IMAGE GENERATION

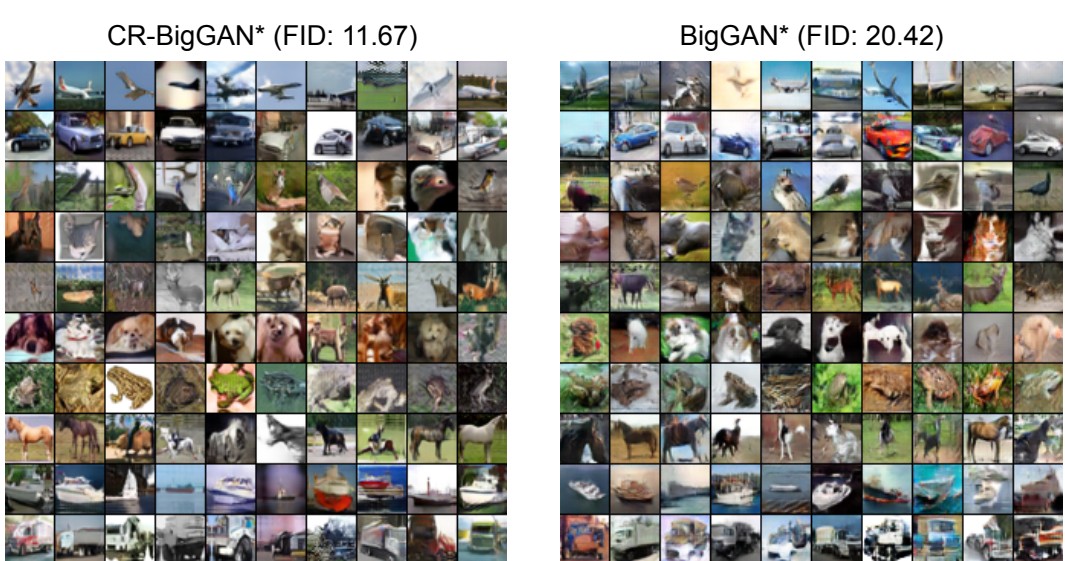

Figure E1: Comparison of generated samples for conditional image generation on CIFAR-10. Each row shows the generated samples of one class.

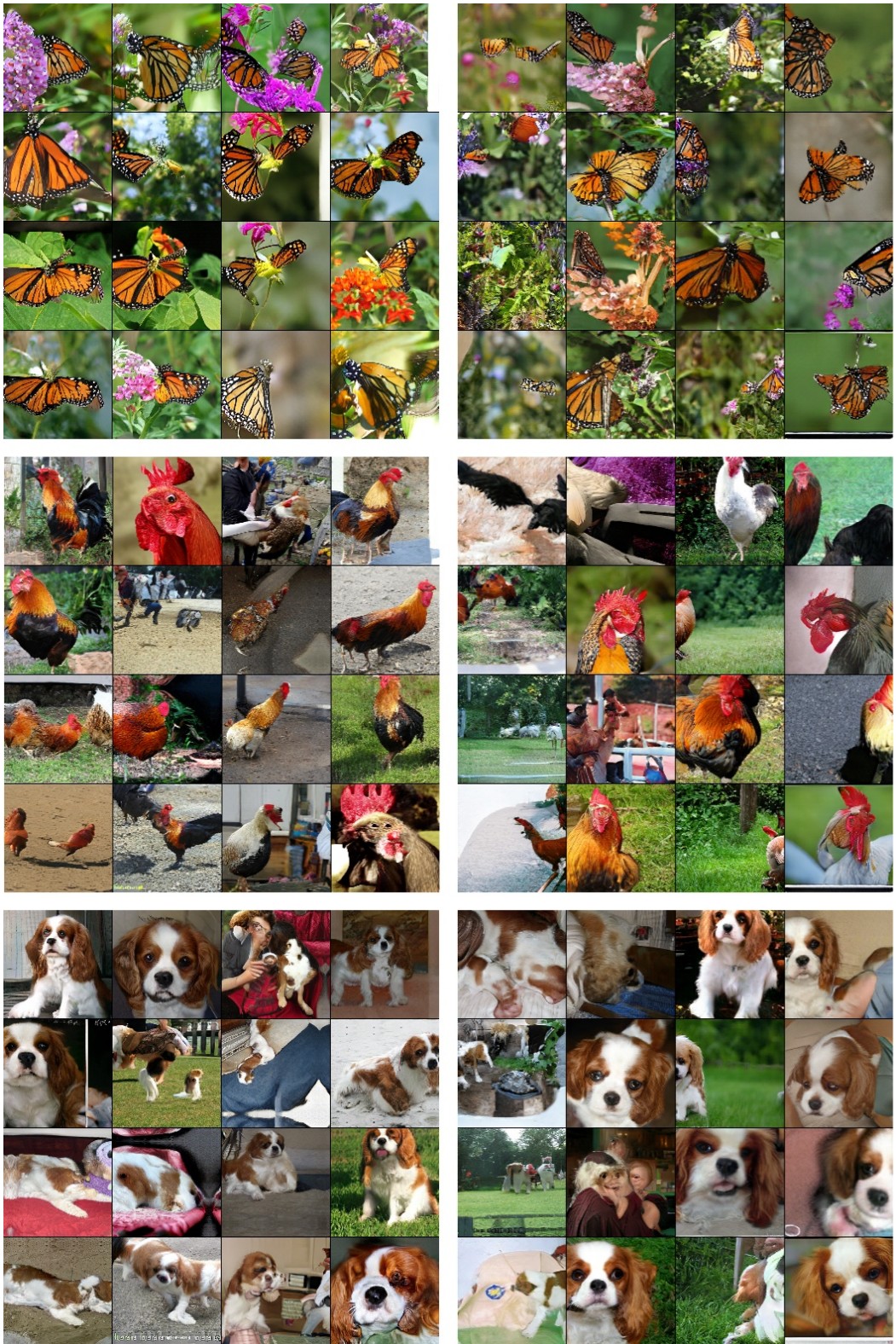

Figure E2: Comparison of conditionally generated samples of BigGAN* and CR-BigGAN* on ImageNet. (**Left**) Generated samples of CR-BigGAN*. (**Right**) Generated samples of BigGAN*.

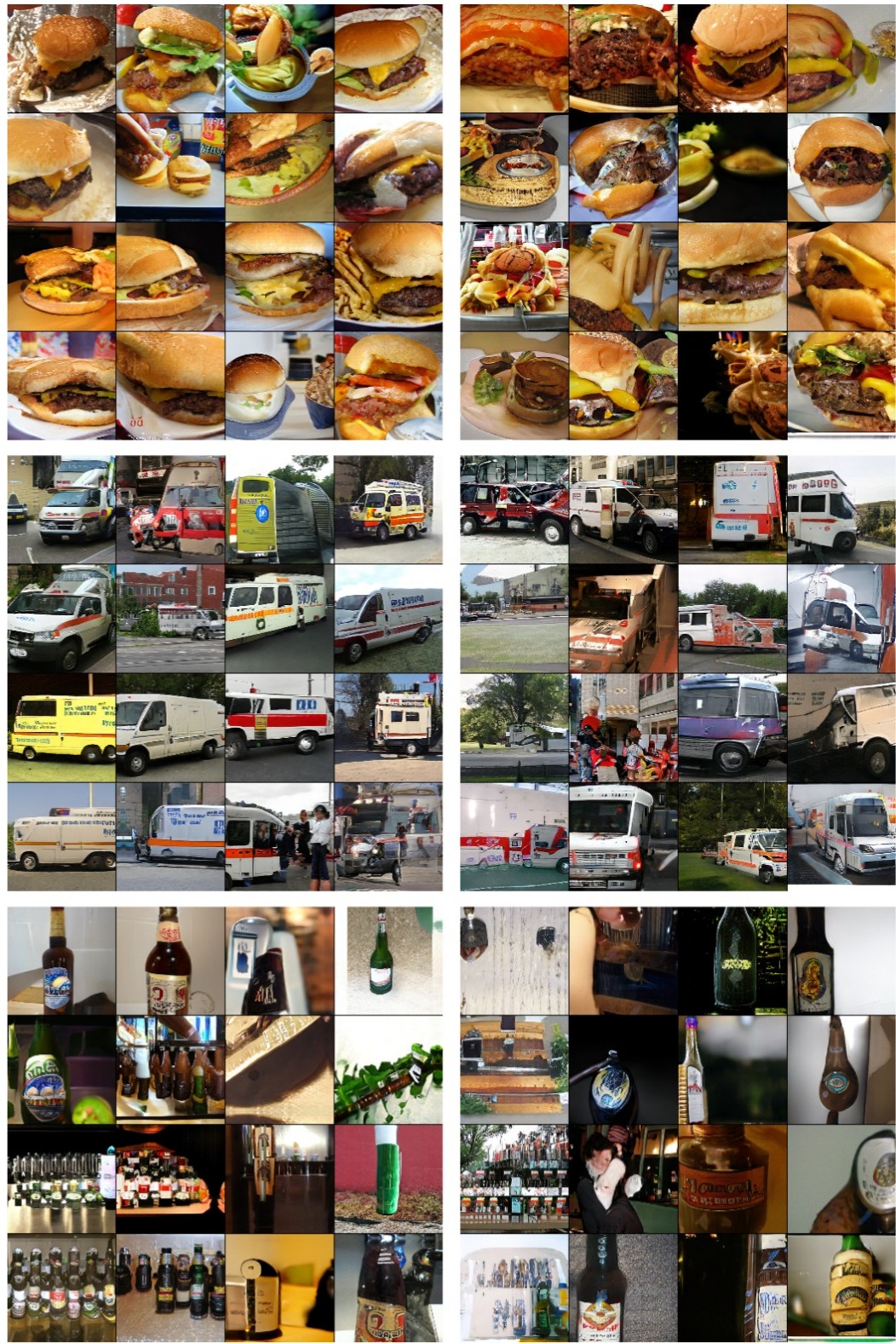

Figure E3: More results for conditionally generated samples of BigGAN* and CR-BigGAN* on ImageNet. (**Left**) Generated samples of CR-BigGAN*. (**Right**) Generated samples of BigGAN*.

# F    COMPARISON WITH INCEPTION SCORE

Inception Score (IS) is another GAN evaluation metric introduced by Salimans et al. (2016). Here, we compare the Inception Score of the unconditional generated samples on CIFAR-10. As shown in Table F1, Figure F1 and Figure F2, consistency regularization achieves the best IS result with both SNDCGAN and ResNet architectures.

| Setting | W/O | GP | DR | JSR | Ours (CR-GAN) |
|---|---|---|---|---|---|
| CIFAR-10 (SNDCGAN) | 7.54 | 7.54 | 7.54 | 7.52 | **7.93** |
| CIFAR-10 (ResNet) | 8.20 | 8.04 | 8.09 | 8.03 | **8.40** |

Table F1: Best Inception Score for unconditional image generation on CIFAR-10.

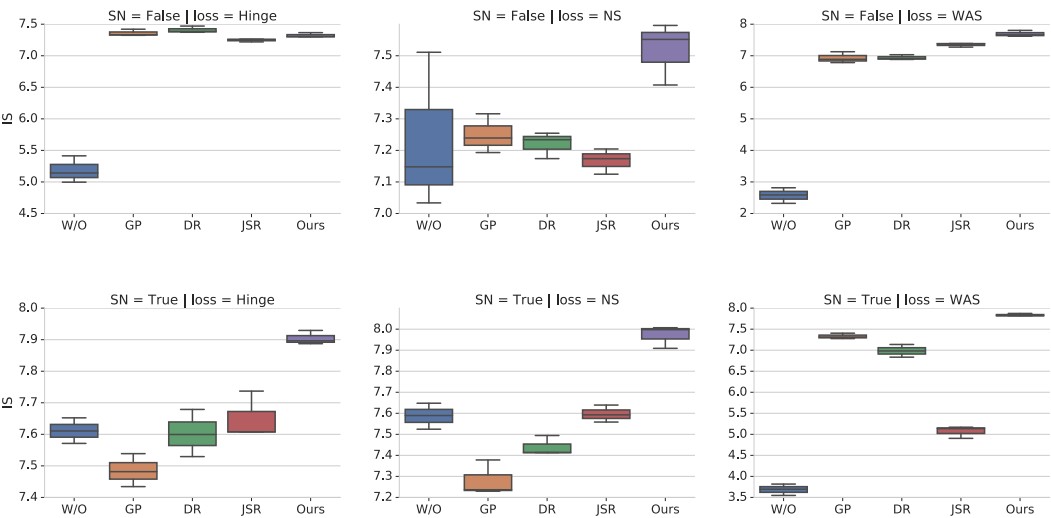

Figure F1: Comparison of IS with a SNDCGAN architecture on different loss settings. Models are trained on CIFAR-10.

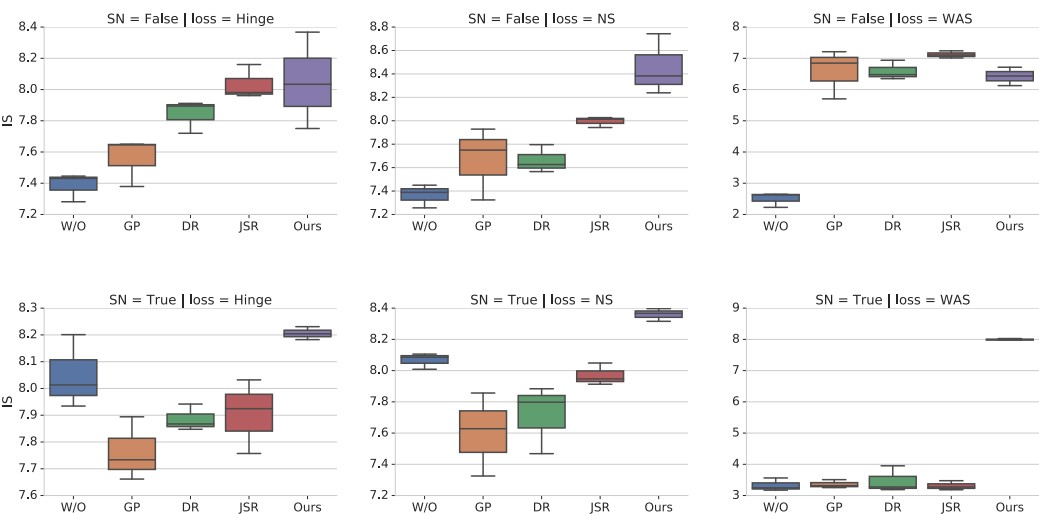

Figure F2: Comparison of IS with a ResNet architecture on different loss settings. Models are trained on CIFAR-10.

## G    EFFECT OF THE NUMBER OF LAYERS REGULARIZED IN DISCRIMINATOR

Here, we examine the effect of the number of layers regularized in discriminator. In this experiment, we use SNDCGAN architecture with NS loss on the CIFAR-10 dataset. There are 8 intermediate layers in the discriminator. To start, we add consistency only to the last layer (0 intermediate layers). Then we gradually enforce consistency for more intermediate layers. We use two weighting variations to combine the consistency loss across different layers. In the first setting, the weight of each layer is the inverse of feature dimension $d_j$ in that layer, which corresponds to $\lambda_j = 1/d_j$ in Equation 4. In the second setting, we give equal weight to each layer, which corresponds to $\lambda_j = 1$. The results for both settings are shown in Figure G1. In both settings, we observe that consistency regularization on the final layer achieves reasonably good results. Adding the consistency to first few layers in the discriminator harms the performance. For simplicity, we only add consistency regularization in the final layer of the discriminator in the rest of our experiments.

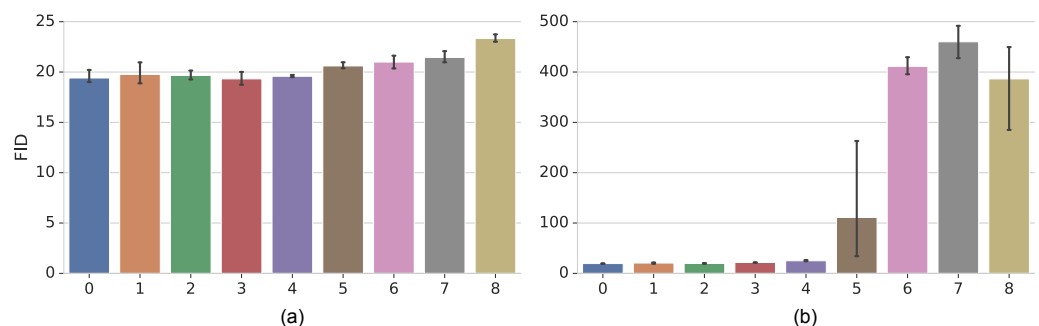

Figure G1: Comparison of consistency regularization on different number of intermediate layers: (a) first weight setting, where the weight for each layer is the inverse of its feature dimension (b) second weight setting, where each layer has equal weight.

## H    CONSISTENCY REGULARIZATION ON THE GENERATED SAMPLES

In this section, we investigate the effect of adding consistency regularization for the generated samples. We compare four settings, no consistency regularization (W/O), regularization only on the real samples (CR-Real), consistency regularization only on the fake samples produced by the generator (CR-Fake) and regularization on both real and fake samples (CR-All). CR-Real is presented in Algorithm 1. CR-Fake has similar computational cost as CR-Real and CR-All doubles the computational cost, since both the augmented real and fakes samples need to be fed into the discriminator to calculate the consistency loss. As shown in Figure H1, CR-Real, CR-Fake and CR-All are always better than the baseline without consistency regularization. In addition, CR-Real is consistently better than CR-Fake. It is interesting to note that CR-All is not always better than CR-real given the extra computational costs and stronger regularization. For example, CR-All improves FID from 20.21 of CR-Real to 15.51 for SNDCGAN, but it also gives slightly worse results for ResNet (14.93 vs 15.07) and for CR-BigGAN* (11.48 vs 12.51). We observe that enforcing additional consistency on the generated samples gives more performance gain when the model capacity is small and that gain decreases when model capacity increases. For computational efficiency and simplicity of the training algorithm, we use consistency regularization on real samples for the rest of our experiments.

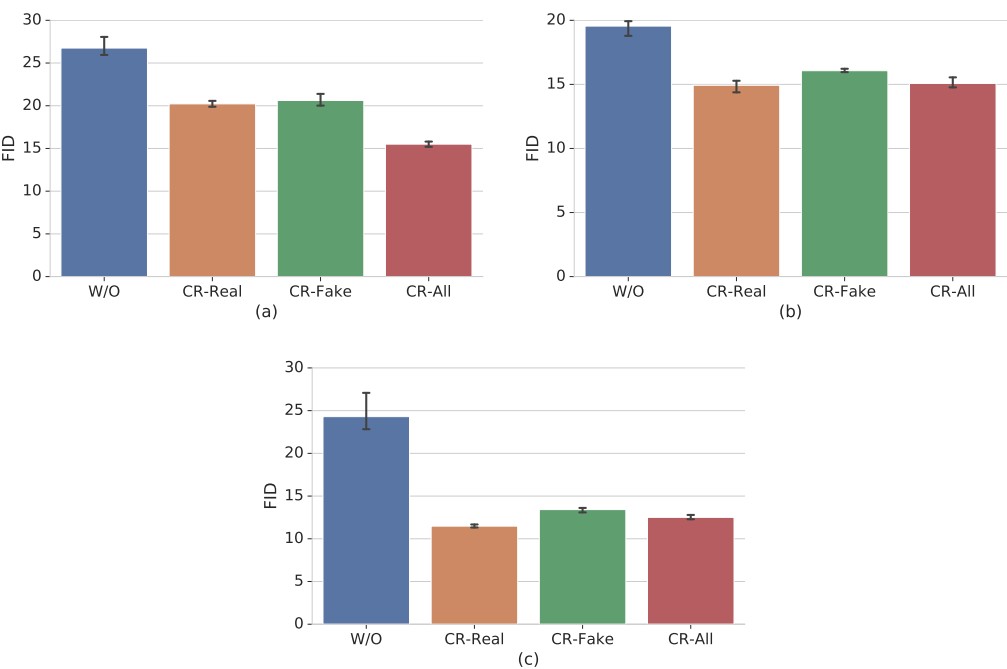

Figure H1: Comparison of FID scores with no consistency regularization (W/O), regularization only on the real samples (CR-Real), consistency regularization only on the fake samples produced by the generator (CR-Fake) and regularization on both real and fake samples (CR-All) for (a) unconditional image generation on CIFAR-10 with SNDCGAN, (b) unconditional image generation on CIFAR-10 with ResNet, (c) conditional image generation on CIFAR-10 with CR-BigGAN*.

