# OpenReview forum: "Consistency Regularization for Generative Adversarial Networks"
_ICLR.cc/2020/Conference — Accept (Poster)_

### Official Review · AnonReviewer1 · 2019-10-28
**Official Blind Review #1**

**Rating:** 6

**Review:**

The topic of this paper is out of the reviewer's domain (Bayesian optimization, RL, and neuroscience). The reviewer has been reviewing ICLR for several years. Such mismatches had not happened in the past.

The reviewer doesn't think this paper reached the bar of a good ICLR paper but hesitates to reject.


This work proposed a training stabilizer for GANs based on the notion of Consistency Regularization. Experimentally, the authors had augmented data passed into the GAN discriminator and penalize the sensitivity of the ultimate layer of the discriminator to these augmentations.

The authors claimed "We conduct a series of ablation studies to demonstrate that the
consistency regularization is compatible with various GAN architectures and loss
functions. Moreover, the proposed simple regularization can consistently improve
these different GANs variants significantly. "












**Experience Assessment:**

I do not know much about this area.

**Review Assessment: Checking Correctness Of Derivations And Theory:**

N/A

**Review Assessment: Checking Correctness Of Experiments:**

I did not assess the experiments.

**Review Assessment: Thoroughness In Paper Reading:**

I made a quick assessment of this paper.

---

> ### Author Response · Authors · 2019-11-15
> **Thanks for your review!**
>
> Thank you for your comments.
>
> In this paper,  we propose a simple, effective, and computationally cheap method – consistency regularization – to improve the performance of GANs. We also have conducted extensive experiments to verify the proposed method. We achieved state of the art results for both conditional and unconditional image generation. Since we have substantially improved the writing of our paper and added more experiments during the rebuttal process, we would be grateful if the reviewer would take another look of the updated version.
>
> See also the other two reviews for unbiased opinions on the merits of our submission.

---

### Official Review · AnonReviewer4 · 2019-10-28
**Official Blind Review #4**

**Rating:** 6

**Review:**

This paper proposes to use Consistency Regularization for training GANs, a technique known to work well in unsupervised learning. The technique consists in applying a transformation to real images and enforcing that the features of the discriminator between the transformed inputs and the original inputs are similar. The author show that using this technique enables them to improve the performance of a standard GAN significantly on CIFAR10. They also carry an ablation study studying the influence of the different part of the proposed technique.

Overall I'm in favor of accepting this paper. The paper is well written, with convincing experiments and an interesting ablation study. However I have several minor issues that I think could greatly improve the paper if addressed.

Minor comments:
- I think an idea which is somewhat related but hasn't been mentioned in the paper, is the idea of adding noise to the input when training GANs [1]. I think this is worth mentioning in the related work.

- Related to the previous point, why penalizing features and not directly output ? What about also trying to classify the transformed images as real ? Also you say that penalizing the last layer, I think including the influence of m (eq 2) in the ablation study would be interesting.

- The authors provide some measure of standard deviation on some experiments but not on all of them. It would be nice to systematically report the standard deviation for every experiments.

- In figure 1 the author make the hypothesis that the discriminator will output very different score to images semantically close together. Did the author verify this hypothesis experimentally ?

- Also why penalizing only the samples from the real distribution and not from the generator ? have you tried both ?

- When the test accuracy of the discriminator is low, it could also be that the discriminator is under-fitting, it would be nice to also report the train accuracy for the discriminator.

- I think the conclusion about the effect of consistency regularization vs data augmentation is a bit vague since consistency regularization has no sense without data-augmentation.

- It's quite interesting but also disappointing that combining transformations doesn't give that much of an improvement. Do the author have any intuition why this is the case ? and why learning them one after the other would work ?

References:
[1] Arjovsky and Bottou. "Towards Principled Methods for Training Generative Adversarial Networks." (ICLR 2017)

**Experience Assessment:**

I have published in this field for several years.

**Review Assessment: Checking Correctness Of Derivations And Theory:**

N/A

**Review Assessment: Checking Correctness Of Experiments:**

I assessed the sensibility of the experiments.

**Review Assessment: Thoroughness In Paper Reading:**

I made a quick assessment of this paper.

---

> ### Author Response · Authors · 2019-11-15
> **Thanks for your review! (Response to Q1-Q6 )**
>
> Thank you for all the valuable comments.
>
> Q1: Related work [1]
>
> Thank you for pointing out the related work. We cited this paper in our revision.
>
> Q2: Regularizing with features vs output
>
> In our method, we penalize sensitivity of the last layer (which is one dimensional) of the discriminator. It is actually the output for both hinge loss and Wasserstein loss. We add the consistency regularization before sigmoid activation for NS loss to be consistent with the other two losses. Since  sigmoid function will squash the range of the output, we would need large regularization coefficient to mitigate this. We also verified this experimentally. On CIFAR-10 with DCGAN structure and NS loss:
>
> Setting                                                                     FID
> CR before sigmoid (\lambda=10)                   19.71±0.28
> CR after sigmoid(\lambda=10)                        22.23±0.85
> CR after sigmoid(\lambda=100)                     19.75±0.24
>
> The reason we did not classify the transformed images as real is that we reasoned that consistency cost is more informative than pure 0 or 1 labels. In other words, classifying with 0/1 loss will treat all real images and their transformed images with the same label as “real” without considering semantics, whereas our consistency cost further enforces learning implicit manifold structure that pulls semantically similar images (original real image and the transformed image) to be closer. We will clarify this further in the revision.
>
> We also added the ablation study for the sensitivity of different layers. Details can be seen in our reply to Q2 of Reviewer1 and Appendix G.
>
> Q3: Measure of standard deviation in experiments
>
> We agree with the reviewer and will update the paper to report the result more systematically . It's also worth mentioning that the box plots in our paper help to show the variance of the experiments.
>
> Q4: Effect of CR regularization on the discriminator output
>
> Yes, we have verified it experimentally. With consistency regularization, the average output distance between real and augmented sample is 0.00449±0.00149. However, without consistency regularization, the average output distance keeps increasing during training.The final average distance is 4.50±1.54, which is roughly around 1000 times larger than the one with consistency regularization.
>
> Q5: Consistency regularization on the images sampled from the generator
>
> Yes, we have tried to add the consistency on the generator outputs as well. In such case,
> the computational cost is doubled but the performance gains vary according to different experiment settings. It  improves FID from 20.21±0.28 to 15.51±0.25 for SNDCGAN, but it also gives slightly worse results for ResNet from 14.93±0.40 to 15.07±0.34 and for CR-BigGAN* from 11.48±0.21 to 12.51±0.21.
> We have added the discussion and more results  in Appendix H.
>
> Q6: Train/test accuracy of discriminator
>
> We have added the training accuracy in Figure 5. For the vanilla GAN, training accuracy of the discriminator is over 0.7, but the test accuracy is around 0.2. It indicates the discriminator is overfitting in such cases.

---

> ### Author Response · Authors · 2019-11-15
> **Thanks for your review (Response to Q7-Q8)**
>
> Thank you for all the valuable comments.
>
> Q7: Consistency regularization vs data augmentation
>
> We are not sure whether we understood your comments clearly, but here we tried to provide our response with our best attempt to interpret your comments. (Apologizes if we misunderstood your question, and in such a case, we will further appreciate if you can clarify us with further comments/question accordingly). To be on the same page, we first want to clarify that, by “data augmentation”, we mean applying transformation to the original real images and treating them as additional data with label “real” for binary classification task in discriminator training. In contrast, consistency regularization uses “augmented data” but uses them differently by enforcing consistency of discriminator output between real image and its transformed image. In Section 4.1, our goal was to investigate whether our improvement of GAN is due to the fact that we reduce the overfitting of discriminator (in terms of better classifying images into real vs fake) or whether consistency regularization provides a special kind of regularization to the discriminator. If the reason is the former, simply applying data augmentation should have already provided similar benefits. However, according to our experiments, it is not the case.
>
> This suggests an interesting interpretation, which is that the mechanism by which the consistency regularization improves GANs is not simply discriminator generalization (in terms of classifying images into real vs fake). We believe that the main reason for the impressive gain from the consistency regularization is due to learning more semantically meaningful representation for the discriminator. More specifically, data augmentation will simply treat all real images and their transformed images with the same label as real without considering semantics, whereas our consistency regularization further enforces learning implicit manifold structure in the discriminator that pulls semantically similar images (i.e., original real image and the transformed image) to be closer in the discriminator representation space. We will clarify this further in the revision.
>
> Q8. Combining different transformations
>
> We have two possible reasons that combining augmentations does not give the best result. First,  combining augmentations can be also considered as adding stronger regularization, and stronger regularization only helps the model performance within a certain range. Second, generator sometimes also generate samples with augmented artifacts (e.g. cutout). If such artifacts do not exist in the real dataset, it might lead to worse FID performance.
>
> To be more clear, the goal of this experiment is to show that not all augmentations are useful for consistency regularization for GANs. We think further study of data augmentation in consistency regularization for GANs will be an interesting direction. For example, we have seen wide studies about data augmentation for image classification [2][3]. However, different from image classification augmentation, we believe the image augmentation in consistency regularization for GANs needs more careful design to make the resulting image not too far away from real data distribution. We will revise the section to make it more clear.
>
> [2] Cubuk, Ekin D., et al. "Autoaugment: Learning augmentation policies from data." arXiv preprint arXiv:1805.09501 (2018).
> [3] Lim, Sungbin, et al. "Fast autoaugment." arXiv preprint arXiv:1905.00397 (2019).

---

### Official Review · AnonReviewer6 · 2019-10-30
**Official Blind Review #6**

**Rating:** 8

**Review:**

Summary:
The paper presents a new regularization technique termed consistency regularization for training GANs. The idea is the following: the authors propose to penalize the sensitivity of the last layer of the discriminator to augmented images. This idea is simple yet efficient: it is easy to implement, a regularization term is gradient-free, and its computation is up to 1.8 times faster than standard gradient-based regularization techniques. The authors tested different augmentation techniques and concluded that simple ones behave better (e.g., shifting and flipping). The experimental results show an impressive gain in FID measure, renewing the current state-of-the-art score for class conditional image generation on CIFAR-10 dataset.

Pros:
The proposed technique is very simple and intuitive; it easy to implement, and it is computationally cheap. The experiments were held for three runs with different random seeds, supporting its consistency.  The paper is overall clearly written and easy to understand.

Cons:
The reported experimental results are held only for BigGAN architecture while not considering different networks to ensure the stability of the proposed regularization. Also, the paper would benefit from a clear experiment description on CelebA dataset (e.g., adding the results to Table 1).

Questions:
-Have you tried other transforms, which potentially keep images on the manifold, including zoom, resize, rotation, brightness adjustment, etc.?
-How the number of layers in $L_{cs}$ (formulas 2-3) affects FID?
-Have you considered an unconditional setting?

Minor comments:
-It would be more convenient if the authors explicitly numerate subplots; e.g., in Figure 2, it is confusing to refer the subplots labeled by (a)-(f) as written in caption.
-Additionally, it would be nice to include say best FID scores over different loss functions (from Figure 2) to Table 1.
- In section 4.3, you wrote that you tried different $\lambda$ values:  {0,1,10, 100}, but Figure 4 does not cover all of them.
- It would be nice to add implementation details (e.g., optimizer, learning rate parameters, steps per discriminator, etc.) for better reproducibility.
-The paper would benefit from illustrations of generated samples.
-Please check the spelling of the penultimate article name in references (Zhai et al., 2019).


**Experience Assessment:**

I have read many papers in this area.

**Review Assessment: Checking Correctness Of Derivations And Theory:**

N/A

**Review Assessment: Checking Correctness Of Experiments:**

I carefully checked the experiments.

**Review Assessment: Thoroughness In Paper Reading:**

N/A

---

> ### Author Response · Authors · 2019-11-15
> **Thanks for your review!**
>
> Thank you for your valuable comments.
>
> Q1: Effect of different types of transformations
>
> We added experiments to examine different augmentations including random flip, shift, zoom, rotation, brightness and cutout on both CIFAR-10 and CelebA dataset with SNDCGAN and NS loss.
> The results are listed below:
>
> Dataset      shift and flip    brightness       zoom            rotate           cutout          gaussian         SBZR*
> CIFAR-10     20.50±0.12    25.83±0.18    30.44±0.36    28.58±0.14    22.52±0.42    36.72±1.77    27.82±0.83
> CelebA        18.84±0.25    26.81±0.61     24.51±0.42    45.06±6.62    24.86±0.33    44.47±0.45   23.80±0.36
> *SBZR means the combination of shift&flip, brightness, zoom and rotate.
>
> From these two datasets, shifting and flipping achieves the best result and adding gaussian noise usually achieves the worst result, which is consistent with our findings in Sec. 4.2. For CIFAR-10 dataset, changing brightness is better than zooming and rotation, whereas for CelebA, zooming is better compared to rotation and changes in brightness. We think the performance for different augmentations depends on data distribution of different datasets. For example, in the CelebA dataset, zooming effect is more natural than rotation, since the face images are quite well aligned.
>
> Q2: Effect of the number of intermediate layers on FID
>
> We added  one more experiment to study the effect of different numbers of intermediate layers in CR-GAN. We add consistency regularization to the last k intermediate layers. We use two weighting variations to combine the consistency loss across different layers.
> In the first setting, the weight of each layer is the inverse of feature dimension in that layer. In the second setting, we give equal weight to each layer. The FID scores for both settings are shown below.
>
> number of intermediate layers     weight setting1          weight setting2
>          k=0                                                 19.41±0.57                 19.48±0.41
>          k=1                                                 19.76±0.88                 20.53±1.01
>          k=2                                                 19.66±0.36                 19.65±0.57
>          k=3                                                 19.31±0.53                 21.57±0.51
>          k=4                                                 19.57±0.09                 25.03±0.95
>          k=5                                                 20.61±0.26                 111.08±107.43
>          k=6                                                 20.99±0.52                411.31±13.88
>          k=7                                                 21.45±0.46                460.83±26.21
>          k=8                                                 23.32±0.30                386.38±72.37
>
> In both settings, we observe that consistency regularization on the final layer (k=0) achieves reasonably good results. In addition, adding the consistency to the first few layers in the discriminator harms the performance. For simplicity, we only add consistency regularization in the final layer of the discriminator for the rest of our experiments. We also add more details in Appendix G.
>
> Q3: Experiments on unconditional setting
>
> We have done experiments for both conditional and unconditional settings. We have updated the paper to make this more clear. In the updated version, Sec 3.2 is for the unconditional setting and Sec 3.3 is for the conditional setting.
>
> Regarding the minor comments:
> Thank you for all these valuable suggestions. We have edited our paper accordingly. For example,
> (1) We labeled the subplot in Figure 2.
> (2) We added the best FID for each method in Table 1.
> (3) We showed the results to cover all the regularization coefficient in Figure 3.
> (4) We added the implementation details in Appendix A
> (5) We added the illustrations of generated samples in Appendix D and E.
> (6) We fixed the typo in the reference.

---

### Public Comment · ~Weihua_Hu1 · 2019-11-04
**Missing reference on the use of data augmentation for consistency training**

I would like to kindly point out that our work [1] also uses consistency training with data augmentation in the context of unsupervised learning. In fact, your Eq. (3) is quite similar to our consistency loss in Eq. (3) of our paper. I think it would be great for you to discuss relation to our work in your paper. Thanks!

Weihua Hu, Takeru Miyato, Seiya Tokui, Eiichi Matsumoto, Masashi Sugiyama.
Learning Discrete Representations via Information Maximizing Self Augmented Training.
International Conference on Machine Learning (ICML2017), 2017.

---

> ### Author Response · Authors · 2019-11-15
> **Thanks for your comments!**
>
> Thank you for the comments.
>
> We would like to point out that consistency regularization is not a new concept. As mentioned in our paper, it has been widely used in semi-supervised learning [1-5] and other domains like discrete representation learning (as in your work, which we will add a citation for).  Eq (3) in your paper uses KL divergence for the consistency loss, while we use mean squared error. However, (as we mention in Sec. 2.2 of our paper), both are the common forms of consistency regularization.
>
> To the best of our knowledge, our work is the first to incorporate consistency regularization into the GAN framework and demonstrate significant improvement over prior state-of-the-art GAN results.
>
> [1] Mehdi Sajjadi, Mehran Javanmardi, and Tolga Tasdizen. Regularization with stochastic transformations and perturbations for deep semi-supervised learning. In NeurIPS, 2016
> [2] Samuli Laine and Timo Aila. Temporal ensembling for semi-supervised learning. arXiv preprint arXiv:1610.02242, 2016.
> [3] Avital Oliver, Augustus Odena, Colin A Raffel, Ekin Dogus Cubuk, and Ian Goodfellow. Realistic evaluation of deep semi-supervised learning algorithms. In NeurIPS, 2018.
> [3] Xiaohua Zhai, Avital Oliver, Alexander Kolesnikov, Lucas Beyer. S4L: Self-Supervised Semi-Supervised Learning. arXiv preprint arXiv:1905.03670, 2019.
> [4] Qizhe Xie, Zihang Dai, Eduard Hovy, Minh-Thang Luong, and Quoc V Le. Unsupervised data augmentation for consistency training. arXiv preprint arXiv:1904.12848, 2019
> [5]David Berthelot, Nicholas Carlini, Ian J. Goodfellow, Nicolas Papernot, Avital Oliver, and Colin Raffel. MixMatch: A holistic approach to semi-supervised learning. arXiv:1905.02249, 2019.

---

### Public Comment · ~Shichang_Tang1 · 2019-12-24
**Dropout in the hidden layers of the Discriminator**

Hi, I would like to point out that [1] uses a consistency regularization term in an effort to enforce the Lipschitz constraint in WGAN.  In their experiments, they find that adding dropout noise in the hidden layers (instead of the input) of the Discriminator for consistency regularization can improve the performance of WGAN-GP.  It would be interesting if you could explore the effect of such augmentations in the intermediate layers as well.

As far as the BigGAN architecture is concerned, [2] finds that "using dropout in D would improve training by reducing its capacity to memorize, but in practice this degrades training". But would BigGAN benefit if dropout is used for consistency regularization?


[1] Wei, Xiang, et al. "Improving the Improved Training of Wasserstein GANs: A Consistency Term and Its Dual Effect." (ICLR 2018).

[2] Brock, Andrew, Jeff Donahue, and Karen Simonyan. "Large scale gan training for high fidelity natural image synthesis." (ICLR 2019).

---

> ### Author Response · Authors · 2020-01-27
> **Thanks for your comments.**
>
> Thanks for your valuable suggestions.
> Adding noise in the intermediate layers to enforce consistency is an interesting direction and we will explore in our future work .
> As you mentioned, [1] is adding dropout in the hidden layers as perturbation.  In this paper, we mainly focus on augmentation on the original data. In this way, we can use prior knowledge to choose some domain specific augmentations to enforce consistency in the data manifold. For example, in the image domain, random flipping and shifting image pixels generally works better and adding gaussian noise on pixels can lead to degraded performance.
> Thanks for pointing out the related work and we will cite [1] in the revision of this paper.

---

### Public Comment · ~Junsoo_Ha1 · 2020-03-02
**Question on the implementation of SNDCGAN**

Hi, I would like to ask a minor question on the implementation of SNDCGAN used for the experiments.

The paper mentions that all the experiments are done with the open-source code from Compare GAN [1], which I have found that they are using larger D (64-128-128-256-256-512-512 conv blocks) [2] than original SNDCGAN architecture  (64-64-128-128-256-256-512 conv blocks) [3]. Could you clarify on the actual detailed architecture ([1] or [2]) used for the experiments?

Thanks!


[1] A Large-Scale Study on Regularization and Normalization in GANs, Karol Kurach, Mario Lučić, Xiaohua Zhai, Marcin Michalski, Sylvain Gelly; ICML 2019
[2] https://github.com/google/compare_gan/blob/19922d3004b675c1a49c4d7515c06f6f75acdcc8/compare_gan/architectures/sndcgan.py#L121
[3] Spectral Normalization for Generative Adversarial Networks, Takeru Miyato, Toshiki Kataoka, Masanori Koyama, Yuichi Yoshida; ICLR 2018

---

### Decision · Program_Chairs · 2019-12-19

**Decision:**

Accept (Poster)

**Comment:**

The paper proposes a simple and effective way to stabilize training by adding consistency term to discriminator. Given the stochastic augmentation procedure $T(x)$ the loss is just a penalty on $D$. The main unsolved question why it help to make discriminator "smoother" in the consistency case for a standard GAN (since typically, no constraints are enforced). Nevertheless, at the moment this a working heuristics that gives new SOTA, and that is the main strength. The reviewer all agree to accept, and so do I.